# Estimating $CO_2$ emissions due to present and future suborbital space tourism industry

Angela M. Huang[1,2], Yangyang Xu[2]*

1 Ronald Reagan High School, San Antonio, Texas, United States of America, 2 Department of Atmospheric Sciences, Texas A&M University, Eller Oceanography & Meteorology Building, College Station, Texas, United States of America

* yangyang.xu@tamu.edu

## Abstract

Estimating and predicting $CO_2$ emissions are the keys to achieving the overall greenhouse gas emission goals for 2030 and 2050. Recent surges in space tourism have generated significant interest and concern. However, the associated $CO_2$ emission and potential future increase remain largely uncertain due to limited publicly available information from the industry. Focusing on suborbital flights, we develop an analytical model to estimate the associated $CO_2$ emissions, considering differences in spacecraft, fuel types, and flight characteristics. The model is based on basic physical principles and fundamental kinetics of the launching and landing phases. We find that suborbital tourism releases 400–1,000 times more $CO_2$ per passenger per hour compared with commercial aviation flights. The spacecraft using liquid hydrogen as fuel releases nearly the least amount of $CO_2$; however, it is still associated with 90–106 metric tons of indirect $CO_2$ emissions due to the fuel production process. In contrast, spacecraft using kerosene (RP-1) and solid fuel (HTPB), respectively, release significantly more. However, if methane (natural gas) is used as fuel, its emissions may become comparable with those using liquid hydrogen, but its maximum reduction in emission can only be less than 10% because that is the amount associated with production. Generally, conventional rocket fuels generate more $CO_2$ than emerging fuels such as hydrogen and methane. Even though the total emissions of suborbital flights are still small compared with commercial aviation, their emission intensity (i.e., per passenger per hour) is enormous (85–226 tons vs. 250 kg). If the demand for suborbital flights increases significantly, as some have speculated, the total annual emission can be as large as 21 Mt $CO_2$ per year in a decade, which exceeds the annual emission of more than 100 countries. Even adopting a conservative estimate, $CO_2$ emissions for suborbital travel can grow to a similar size to that of Congo (with a population of nearly 100 million). Therefore, the $CO_2$ emission from emerging suborbital travel needs more attention in the future.

**Data availability statement:** All relevant data are within the paper and its Supporting Information files.

**Funding:** The author(s) received no specific funding for this work.

**Competing interests:** The authors have declared that no competing interests exist.

## Introduction

Understanding, estimating, and predicting greenhouse gas (GHG) emissions, particularly carbon dioxide ($CO_2$) emissions, are keys to addressing climate change. Consequently, $CO_2$ emissions from different economic sectors have been studied and assessed as a dominant climatic factor [1–4]. Transportation emits a significant amount of $CO_2$ each year, which accounts for nearly 1/3 of global greenhouse gas emissions [5–7]. Identifying high-emission components of transportation (such as aviation) and appropriately addressing them would be particularly effective in reducing $CO_2$ emissions [8]. Many aviation-related emissions, such as carbon dioxide ($CO_2$), nitrogen oxides ($NO_x$), aerosols (e.g., soot and sulfate), and vapor-induced contrails, contribute to the radiative forcing impacting global climate, with $CO_2$ being the dominant factor [9–11]. $CO_2$ emissions from aircraft were approximately 0.5 Gt/year in the early 1990s, which accounted for about 2% of total anthropogenic $CO_2$ emissions or about 13% of transportation [12]. In 2019, total $CO_2$ emissions from aviation reached approximately 0.9 Gt [13]. If this 4%/year growth rate continues, $CO_2$ emissions would reach 3.3 Gt/year in 2050. The aviation-related emissions and their impacts on climate change have been extensively studied [9,12,14]. However, travel by private jets has recently garnered public attention [15,16], which can emit at least ten times more pollutants than commercial planes per passenger [15,17]. Transport & Environment [17] showed that private jet travel increased 31% per year from 2015 to 2019 and Collins, Ocampo [15] found that COVID-19 greatly boosted the demand for private jet travel in 2020. Moreover, there has also been a renewed interest in commercial supersonic flights [18,19], which are expected to consume seven to nine times more fuel per passenger than current commercial flights [20].

In contrast to the aviation sector, the climatic effects of space flights, many of which are now increasingly related to tourism, have only recently begun to gain attention [21]. Space tourism refers to human space travel for recreational purposes, including planetary, lunar, orbital, and suborbital travels [22–25]. Among these types, suborbital tourism, which reaches an altitude of around 100 km (the Kármán line, the conventional boundary between Earth's atmosphere and outer space), is more imminent due to its lower technical and economic barriers and greater market reach. For example, orbital flights and a week-long visit to the International Space Station (ISS) cost tens of millions of dollars per person per trip [26]. Commercial lunar flights are projected to begin in the 2040s [27,28] and travel to Mars in the 2050s [29]. However, suborbital flights are much more accessible, with costs of around $200,000 to $400,000 per person, so they have become the primary target market in recent years [30]. Several countries and companies have joined the competition. The U.S. has successfully completed a few commercial suborbital trips, and China aims for its first flight in 2025, though delays are likely. Various policies and regulations on operational safety, medical requirements, and emergency contingency have been established, which can facilitate a wider public acceptance of space tourism [28,31,32].

Given the recent surge in suborbital tourism and its potential future growth, a thorough quantification of $CO_2$ emissions associated with suborbital tourism is

needed. Rocket and spacecraft launches are known to have high emission intensity, generating 18–100 times more $CO_2$ per passenger compared with a typical long-haul aircraft flight by some estimates [33–35]. Previous space-related climatic and environmental studies [35–41] provide a useful scientific basis but cannot be directly applied to suborbital flights. For instance, FAA [37] focused on horizontally launched rockets, which is not the primary launch method used for suborbital spacecraft. Ross, Toohey [40], Ross and Sheaffer [41], and Maloney, Portmann [39] studied the ozone depletion effects of traditional rocket fuels that release black carbon using observations and simulations. Larson, Portmann [38] studied a two-phase launch rocket called Skylon, but a single-phase launch is primarily used by suborbital tourism spacecraft nowadays.

Suborbital flights and their spacecraft differ significantly from each other in terms of fuel types, fuel efficiency, engines, flight characteristics, etc. Very few studies on suborbital GHG emissions have relied on indirect information, leading to significant uncertainty and potential disputes. For example, Virgin Galactic claimed that its carbon footprint per passenger was comparable to a business-class round trip from London to New York; however, the Financial Times reported that it emitted 1,238 kg of $CO_2$ per passenger [42] – 60 times higher. Fawkes [43] provided an even higher estimate for Virgin Galactic's SpaceShipTwo at 3,113 kg per passenger. Even greater disputes arise over the estimates of non-traditional aviation fuels, such as liquid hydrogen. The World Inequality Report argued that an 11-minute suborbital joyride by Blue Origin created more carbon emissions than 1 billion people produced in their entire lifetime [44]. The claim received widespread media coverage but was questioned in terms of how the carbon footprint of liquid hydrogen was estimated. Rigorous and independent studies should be conducted to quantify $CO_2$ emissions associated with suborbital travel and the indirect carbon footprint of fuel production [43]. The latter is particularly relevant to hydrogen as it does not directly emit $CO_2$ when burned.

There is a severe lack of publicly available information on the emission intensity of spacecraft and rocket engines, particularly regarding non-traditional fuels. In this study, we aim to estimate $CO_2$ emissions associated with suborbital flights (including production and combustion processes) both for the present and the future by (1) developing a generalizable analytical model based on scientific principles (Newton's law and aerodynamics) to link the suborbital flight process (launch, cruise, and landing) with kinetic energy requirement and fuel consumption, (2) collecting data on various spacecraft and fuels used by the major carriers to estimate their $CO_2$ emissions per flight, as well as emission intensity (i.e., per passenger per hour), and (3) projecting future $CO_2$ emissions for the suborbital space tourism industry based on published growth scenarios.

## Background of suborbital tourism

The U.S. has dominated suborbital flights since Alan B. Shepard completed the first American suborbital flight in 1961. Scaled Composite's SpaceShipOne performed the first commercial suborbital flight in 2004. Since then, private companies such as Virgin Galactic, Armadillo Aerospace, Blue Origin, and Masten Space Systems have launched or are working to launch suborbital flights (Table 1). EADS Astrium, a subsidiary of Airbus, announced its space tourism project in 2007. Armadillo Aerospace, based in Texas (now rebranded as Exos Aerospace), had been developing a spacecraft called Hyperion but has not launched, and its current status is unclear. XCOR Aerospace was developing a suborbital vehicle called Lynx, which was halted in May 2016; XCOR later filed for bankruptcy in 2017 due to financial reasons.

Despite many pioneering efforts, the main or imminent players in the industry today are the so-called Big-3 (i.e., Virgin Galactic, Blue Origin and SpaceX), which we briefly introduce below.

(1) Virgin Galactic, founded in 2005, started this endeavor by SpaceShipTwo-class spacecraft. The first of these spacecraft, *VSS Enterprise*, was intended to commence its first commercial flights in 2015; however, it crashed in 2014 during a test flight in the Mojave Desert [46]. A second spacecraft, *VSS Unity*, completed a successful test flight on July 11, 2021, and its first commercial spaceflight on June 29, 2023, with 6 crew members on board. As of the second quarter of 2024, Virgin Galactic had more than 800 reservations for upcoming flights.

**Table 1. Status of space tourism companies [30,45–48].**

| Company | Project | Launch type | Propellant | Target destination | Seats | Trip duration | Features | Status |
|---|---|---|---|---|---|---|---|---|
| Benson Space Company | X-1 | Vertical take-off, horizontal landing | Hybrid (rubber and nitrous oxide) | n/a | n/a | n/a | Low g-force | Abolished in 2008 |
| Rocketplane Kistler | K-1/XP | Vertical | RP-1 and liquid oxygen | Orbit | 5 | n/a | Fully reusable orbital system | Bankrupted in 2010 |
| Xcor Aerospace | Xerus | Horizontal | RP-1 and liquid oxygen | Suborbital | 2 | n/a | Rocket motor for take-off | Bankrupted in 2017 |
| Armadillo Aerospace | Pixel | Vertical | Ethanol and liquid oxygen | Lunar | n/a | n/a | Modern computer control | unknown |
| Interorbital System | Neptune | Vertical | Hypergolic hydrocarbon | Orbit | 6 | n/a | Two-stage launch, and fuel does not need ignition system | In testing |
| EADS Astrium | Astrium Spaceplane | Horizontal | Methane and liquid oxygen | Suborbital | 5 | n/a | Jet engine for take-off and rocket to reach space | In test and look for investors |
| Space Adventures | Soyuz MS-20 | Vertical | RP-1 and liquid oxygen | Orbital and to the ISS | 3-4 | 8-14 days | First paid space tour | One of the earliest private spaceflight companies. |
| Blue Origin | New Shepard | Vertical | Liquid hydrogen and oxygen | Suborbital | 4-6 | 11 minutes | Commercialized | Completed 25 manned flights as of May 2024 |
| Virgin Galactic | Space-ShipTwo | Horizontal | Hybrid (rubber/HTPB and nitrous oxide) | Suborbital | 8 | 1.5 hours | First controlled descend of civilian spacecraft (SpaceShipOne)* | Completed 12 manned flights as of June 2024 |
| SpaceX | Falcon/Dragon | Vertical | RP-1 and liquid oxygen | Orbital, potentially suborbital, | 7 | n/a | Flight to the highest altitude for private spacecraft | Private orbital travel completed with spacewalk. Suborbital status unknown. |
| SpaceX | Starship | Vertical | Methane and liquid oxygen | Potentially from suborbital to Mars | TBA | n/a | It is the largest and most powerful spacecraft ever to fly. | A few launches, reaching the highest altitude of nearly 200 km |

*SpaceShipOne and SpaceShipTwo systems were designed and built by Scaled Composite which is a subsidiary of Northrop Grumman. Virgin Galactic is the carrier.

(2) Blue Origin developed the *New Shepard* (NS) reusable suborbital launch system specifically to enable short-duration space tourism for a maximum of six passengers. The rocket successfully launched with four passengers on July 20, 2021, including Jeff and Mark Bezos, reaching an altitude of 107 km (66 mi).

(3) In contrast, SpaceX has not formally launched suborbital flights for tourism yet. However, SpaceX has strong technological readiness for developing vehicles for the International Space Station (Falcon/Dragon), Moon (Falcon 9, Falcon Heavy, Starship), and Mars (Starship). Indeed, SpaceX completed the first orbital spacewalk by civilians in September 2024 [49]. Moreover, SpaceX has envisioned a suborbital transportation system using Starship to go from New York to Shanghai in 39 min [50,51]. In this paper, to compare with other main carriers, we limit our analysis to the suborbital travels, for which Falcon/Dragon can be primary vehicles, to make it comparable with the other two carriers. Starship is much larger and heavier and can be considered for future studies.

The so-called Big-3 is representative of the suborbital tourism industry due to their current market dominance and long-term commercial plans. In addition, the Big-3 use different fuels for their spacecraft; therefore, their emissions can be representative of different scenarios of the suborbital tourism industry.

## Model

### Kinetics of suborbital flights

Typically, the suborbital flight launches from the ground, reaches an altitude of 100 km above the Earth's surface (i.e., Kármán line), falls back right away or cruises for a short period and then falls back in a controlled manner via rocket propelling (i.e., no parachutes), as shown in Fig. 1a. Suborbital tourism market is expected to expand significantly and will dominate space tourism in the future (Fig. 1b).

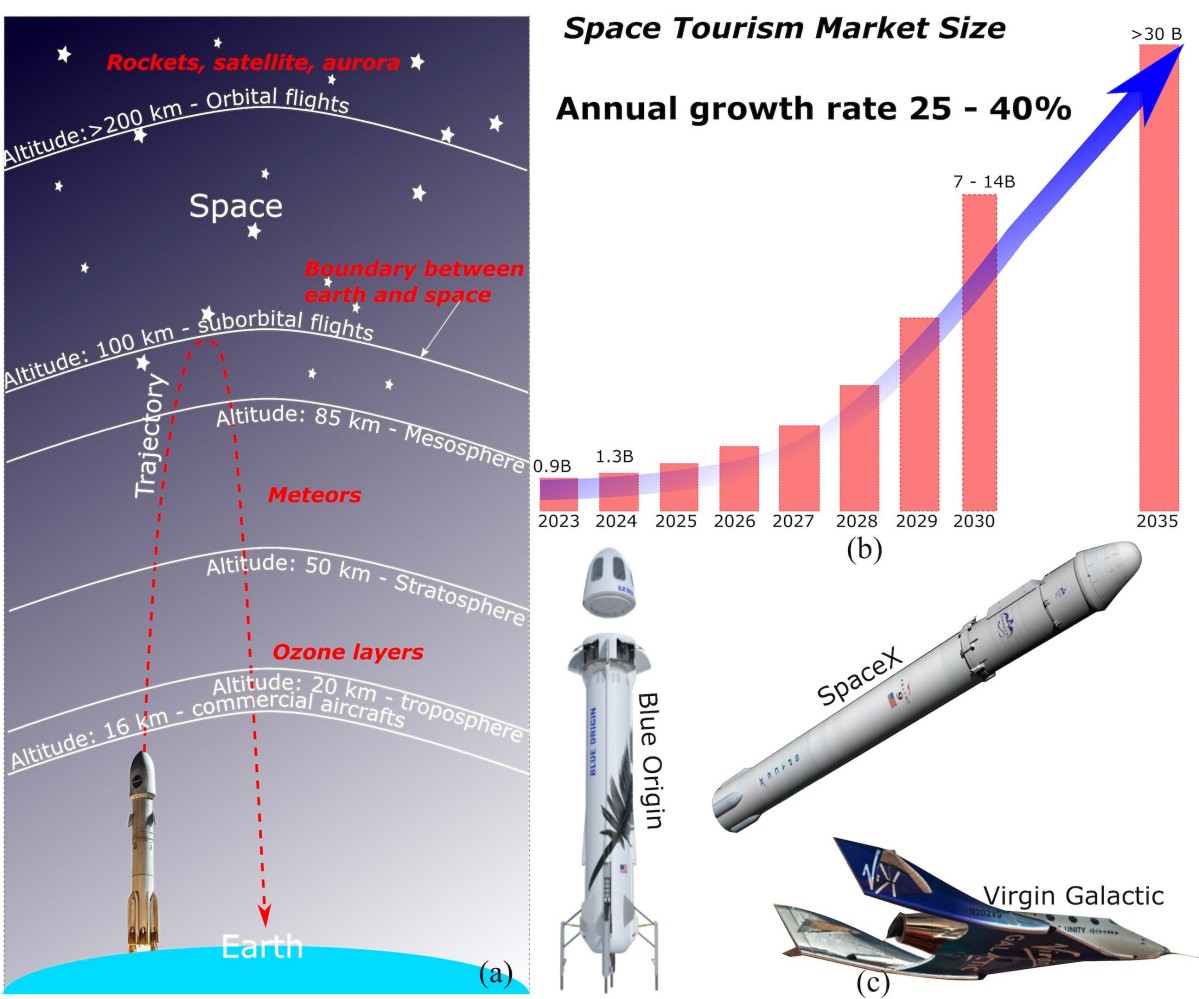

**Fig. 1. Suborbital flights: (a) Commercial aviation, suborbital and orbital altitudes, (b) space tourism market and prediction (data from different sources: Space Tourism Industry Research Report, Grand View Research, Global Mart Insights), (c) major carriers of suborbital tourism: Virgin Galactic (VSS), Blue Origin (NS), and SpaceX (Dragon).**

Based on Newton's second law, the motion of the spacecraft can be described as Eq. 1:

$$M(t) \cdot a(t) = F_n(t) - M(t) \cdot g - F_{ad} \tag{1}$$

where $M(t)$ is the total mass of the spacecraft, which varies with time due to propellant consumption. $a(t)$ is acceleration. The acceleration of gravity, $g$, is a function of altitude, but it is only 3% smaller at the altitude of 100 km than at sea level, so it is treated as a constant here. $F_n$ is the net thrust, the key quantity we later need to estimate based on rocket types and fuel consumption rates. $F_{ad}$ is the air drag force, which is a function of drag coefficient ($C_d$), air density ($\rho$), flow velocity ($v$), and reference area ($A$), as shown in Eq. 2:

$$F_{ad} = 0.5 C_d \rho v^2 A \tag{2}$$

Flow velocity, $v$, is the relative velocity between the spacecraft and air, which is approximated by the vertical velocity of the spacecraft since most suborbital tourism involves a return to the launch site. The drag coefficient, $C_d$, is spacecraft-specific, depending on shape, surface roughness, and fluid type, which usually must be determined by wind tunnel tests. A flat plate has a $C_d$ of 1.28, a wedge-shaped prism with the wedge facing downstream has a $C_d$ of 1.14, a sphere has a $C_d$ that varies from 0.07 to 0.5, a bullet with a $C_d$ of 0.295, and a typical airfoil with a $C_d$ of 0.045. NASA used 0.75 for its model rocket [52], which is adopted for this study.

The reference area, $A$, is the frontal area of the spacecraft. The existing suborbital spacecrafts from different carriers have a similar reference area. Specifically, for Blue Origin's NS-13 and SpaceX's Falcon 9 with the Dragon capsule, the spacecrafts have a cylindrical shape with a diameter of 3.7 m. The spacecraft of Virgin Galactic (VSS Utility) is more airplane-like, with a total width of 8.3 m (including wings) and a cabin diameter of 2.3 m. When converted into an equivalent circle shape, its diameter will be about 3.7 m as well. Thus, the reference area for all three major spacecraft is approximately 11 $m^2$ in this study. The spacecrafts of Blue Origin, Virgin Galactic, and SpaceX (Big-3) are shown in Fig. 1c for illustration. SpaceX Falcon 9 with the Dragon capsule is longer than the other two because it is equipped with a two-stage launching capacity to allow it to go further into orbit and the International Space Station (ISS).

In contrast, Starship of SpaceX is not aimed for tourism purpose but is designed for commercial flights, such as its earth-to-earth flights, to take hundreds of passengers each time, so it is much larger. Its cross-section area is nearly 6 times of Falcon 9 of SpaceX and NS-13 of Blue Origin. Due to its great mass, its launching process is also more complicated. However, its flight may be within the suborbital range, which has great potential of becoming another important source of $CO_2$ emissions in our atmosphere. It deserves another separate and comprehensive study.

The air density, $\rho$, is a function of altitude, i.e., at sea level, it is 1.22 $kg/m^3$ and becomes negligible above 60 km. Based on the ideal gas law, the air density from 0–20 km can be estimated from Eq. 3(a), while the air density from 20–60 km can be estimated using linear interpretation of the monitoring data [53], as shown in Eq. 3(b).

$$\rho = \frac{p_0 M_m}{R T_0} \left(1 - \frac{Lh}{T_0}\right)^{\frac{gM_m}{RL} - 1} \quad (h \leq 20\,km) \tag{3a}$$

$$\rho = -2.2 \times 10^{-6} h + 0.1331 \quad (20 < h < 60\,km) \tag{3b}$$

$$\rho = 0 \quad (h \geq 60\,km) \tag{3c}$$

where $p_o$ and $T_o$ are the pressure and temperature at the sea level, respectively; $M_m$ is molar mass, R is the ideal gas constant, L is the temperature lapse rate (6.5 K/km), and h is the altitude. Eq. 3 is used to only calculate air density by travel segments, and above 60 km air density is treated as 0 because the air density beyond 60 km is less than 0.09% of the value at the sea level [53].

Based on the air density, the air drag force, $F_{ad}$, can thus be expanded into Eq. 4:

$$F_{ad} = 0.375 * \frac{p_0 M_m}{RT_0}\left(1 - \frac{Lh}{T_0}\right)^{\frac{gM_m}{RL} - 1} * v^2 A v^2 A \quad (h \leq 20\text{km}) \tag{4a}$$

$$F_{ad} = 0.375 * \left(-2.2 \times 10^{-6} h + 0.1331\right) * v^2 A \quad (20 < h < 60\text{km}) \tag{4b}$$

$$F_{ad} = 0 \quad (h \geq 60\text{km}) \tag{4c}$$

Combining Eqns. 1 and 4, the required net thrust for the spacecraft at any time can be expressed by Eq. 5:

$$F_n(t) = M(t)\,(a(t) + g) + 0.375\frac{p_0 M_m}{RT_0}\left(1 - \frac{Lh}{T_0}\right)^{\frac{gM_m}{RL} - 1} v^2 A \quad (h \leq 20\text{km}) \tag{5a}$$

$$F_n(t) = M(t)\,(a(t) + g) + 0.375\left(-2.2 \times 10^{-6} h + 0.1331\right) v^2 A \quad (20 < h < 60\text{km}) \tag{5b}$$

$$F_n(t) = M(t)\,(a(t) + g) \quad (h \geq 60 \text{ km}) \tag{5c}$$

Since the velocity (v) and altitude (h) are functions of time (t) and acceleration (a), Eq. 5 can further be written in an integral form as a function of time (t) and acceleration (a(t)), with t, a, v, h being zero as the initial conditions (Eq. 6).

$$F_n(t) = M(t)\,(a(t) + g) + 0.375\frac{p_0 M_m}{RT_0}\left(1 - \frac{L\int_0^t\left(\int_0^t a(t)\right)dt}{T_0}\right)^{\frac{gM_m}{RL} - 1}\left(\int_0^t a(t)dt\right)^2 A \quad (h \leq 20\text{km}) \tag{6a}$$

$$F_n(t) = M(t)\,(a(t) + g) + 0.375\left(-2.2 \times 10^{-6}\int_0^t\left(\int_0^t a(t)\right)dt + 0.1331\right)\left(\int_0^t a(t)dt\right)^2 A \quad (20 < h < 60 \text{ km}) \tag{6b}$$

$$F_n(t) = M(t)\,(a(t) + g) \quad (h \geq 60 \text{ km}) \tag{6c}$$

This becomes our governing equation to estimate the thrust demand, which must be balanced by the thrust supply to be calculated next (Section 3.2).

### Kinetic energy generation

In addition to abiding by a general physics principle, i.e., Newton's law of motion, spacecraft also must follow the engineering principle of a thrust system during flights. To move any object, a propulsion system must provide sufficient thrust, which is the mechanical force generated due to the momentum change of working fluids (high-temperature and high-velocity gases in our case). The governing equation for the generation of net thrust, $F_n$, is given by Eq. 7 below:

$$F_n = \dot{m}v_e + A_e(p_e - p_{am}) \tag{7}$$

where $\dot{m}$ is the exhaust gas mass flow rate (mass flux in unit time), $v_e$ is the exhaust flow velocity, $A_e$ is the flow area at the nozzle, $p_e$ is the pressure at the nozzle, and $p_{am}$ is the (typically lower) ambient pressure. Since the pressure-area term is very small relative to the first term for rocket engines, Eq. 7 can be simplified into Eq. 8 [54].

$$F_n = \dot{m}v_e \tag{8}$$

The velocity of the exiting exhaust gases ($v_e$) can be calculated using Eq. 9 [55].

$$v_e = \sqrt{\frac{TR}{M}\frac{2\gamma}{\gamma-1}\left[1-\left(\frac{p_e}{p_c}\right)^{\frac{\gamma-1}{\gamma}}\right]} \tag{9}$$

where $v_e$ is exhaust velocity at the nozzle, T is the absolute temperature of inlet gas, R is the ideal gas law constant, M is the gas molecular mass (also known as the molecular weight), $\gamma$ is the isentropic expansion factor (i.e., $c_p/c_v$, where $c_p$ and $c_v$ are the specific heats of the gas at constant pressure and constant volume, respectively) and is larger than 1; $p_e$ is the absolute pressure of exhaust gas at the nozzle exit, $p_c$ is the absolute pressure of inlet gas (i.e., chamber pressure), which is larger than $p_e$ and thus $v_e$ is positive.

Combining Eqns. 8 and 9, the generated net thrust, $F_n$, can be calculated from Eq. 10.

$$F_n = \dot{m}\sqrt{\frac{TR}{M}\frac{2\gamma}{\gamma-1}\left[1-\left(\frac{p_e}{p_c}\right)^{\frac{\gamma-1}{\gamma}}\right]} \tag{10}$$

Eq. 10 indicates that net thrust, $F_n$, is a function of chamber gas temperature (T), chamber pressure $p_c$, and exhaust gas pressure ($p_e$). Conceptually, Eq. 10 can be thought of as Eq. 11.

$$F_n = F_{n,0} \cdot f_T \cdot f_{pc} \cdot f_{pe} \tag{11}$$

where $F_n$ at vacuum condition ($F_{n,0}$) is used as the baseline, and other factors are added to calculate the thrust when operating at actual atmospheric conditions (e.g., at the sea level). The modification factors of $f_T$, $f_{pc}$, and $f_{pe}$ incorporate effects of inlet/chamber gas temperature (T), inlet/chamber pressure ($p_c$, very high and generated from the combustion in the chamber), and the pressure of the high-speed exhaust gas ($p_e$, lower at the nozzle), respectively.

The modification factors for inlet/chamber gas temperature ($f_T$) and inlet/chamber gas pressure ($f_{pc}$) can be approximately treated as 1 here since inlet/chamber gas temperature and inlet/chamber gas pressure do not change with ambient pressure significantly, when the ambient pressure deviates from the vacuum condition.

In addition, the exhaust velocity ($v_e$) is insensitive to changes in inlet/chamber gas pressure ($p_c$). According to Huzel and Huang [56], the velocity only increases by 1% when the inlet/chamber pressure ($p_c$) increases by 11%. However, as shown in Eq. 9, the exhaust velocity ($v_e$) is noticeably impacted by the exhaust pressure ($p_e$). Table 2 consistently shows that across various engines, there can be a 14–15% decrease in exhaust velocity when operating with 1 atm exhaust pressure compared to vacuum conditions. Since the thrust is linearly proportional to exhaust velocity (Eq. 7), this means that for the same rate of fuel burning ($\dot{m}$), a spacecraft receives about 14% less thrust at 1 atm than the vacuum condition

**Table 2. The exhaust velocity of rockets using different rocket fuels [56,59,60].**

| | Fuels | $H_2$ | $CH_4$ | RP-1 | HTPB |
|---|---|---|---|---|---|
| Exhaust velocity, $v_e$ (m/s) | $p_e$ at 1 atm | 3816 - 4036 | 3127 - 3484 | 2941 - 3424 | n/a |
| | $p_e$ at vacuum | 4461 - 4697 | 3692 - 4131 | 3510 - 4021 | n/a |
| | Difference | 14.3% | 15.4% | 15.0% | n/a |

if liquid hydrogen is used and receives about 15% less thrust if other fuels are used. Thus, we can approximate $F_n$ from the baseline $F_{n,0}$ using Eq. 12.

$$F_n = (1 - R_r p_e) * F_{n,0} \tag{12}$$

where $R_r$ is velocity reduction factor considering the difference between 1 atm and vacuum condition, which is 0.14 for liquid hydrogen and 0.15 for other fuels. $p_e$ is exhaust pressure in the unit of standard atmospheric pressure (1 atm = 101 kPa); later we adopt $p_{am}$ of ambient pressure, because they are usually close in value (hence the negligible 2nd term in Eq. 7).

Even though Eq. 10 is the rigorous, closed-form function to calculate thrust, $F_n$, using T, $p_c$, and $p_e$; many studies have used Eq. 11 to estimate $F_n$ by considering its deviation from the baseline condition. In practice, the engine is tested for two conditions: at low-pressure conditions (i.e., close to vacuum) and at sea-level pressure (1 atm). The results are used as a reference for further assessment or estimation [57,58]. Here, we adopt this indirect approach to estimate thrust from its baseline performance using Eq. 12.

### Fuel consumption estimates using thrust specific fuel consumption (TSFC)

The combustion of propellants generates thrust to lift a spacecraft. To quantify the engine efficiency, Thrust Specific Fuel Consumption (TSFC) has been defined as the mass of fuel burned per unit time divided by the thrust produced [61]. A higher TSFC indicates greater fuel consumption per unit thrust generation, or, equivalently, lower thrust per unit fuel burning rate. This is expressed in Eq. 13.

$$TSFC = \frac{\dot{m}_f}{F_{n,o}} \tag{13}$$

where $\dot{m}_f$ is the mass flow rate of the fuel, which is the amount of fuel consumed in unit time. TSFCs of various rocket engines range from 200 to 500 g/(kN·s).

Combining Eqns. 12 and 13, the fuel consumption rate, $\dot{m}_f$, at a given time can be calculated in Eq. 14:

$$\dot{m}_f = TSFC \frac{F_n(t)}{1 - R_r p_e} \tag{14}$$

The exhaust pressure, $p_e$, which decreases at a higher altitude, is often estimated based on an exponential function as in Eq. 15 [53].

$$p_e = (1 - 2.256 \times 10^{-5} \cdot h)^{5.26} \tag{15}$$

where $p_e$ is exhaust pressure in atm, and h is the altitude in m.

### Total propellant consumption during the flight and $CO_2$ emissions

In the previous subsections, we separately provided the governing equation (Section 3.1), the estimate of air drag (Section 3.1), thrust (Sections 3.1 and 3.2) and fuel consumption rate (Section 3.3). Therefore, we are now in a position to derive the propellant consumption over the entire flight duration, which will then cumulatively yield total propellant mass consumed. It is noteworthy that the mass of the spacecraft, M(t) in Eq. (6), will decrease due to propellant consumption (Eq. 14); thus, the mass of the spacecraft equals the initial total mass (pre-launch) minus the mass of the consumed propellant (Eq. 16; where $M_0$ is the initial total mass of the spacecraft).

$$M(t) = M_0 - \int_0^t \dot{m}_f \, dt \tag{16}$$

The developed analytical framework becomes Eq. 17 after combining Eqns. 6, 14, and 16 and considering air drag becomes zero beyond 60 km.

$$F_n(t) = \left(M_0 - \int_0^t \left(TSFC\frac{F_n(t)}{1 - R_r p_e}\right) dt\right) \cdot (a + g) + 0.375\frac{p_0 M_m}{RT_0} \cdot \left(1 - \frac{L\int_0^t \left(\int_0^t a\, dt\right) dt}{T_0}\right)^{\frac{gM_m}{RL} - 1} \left(\int_0^t a\, dt\right)^2 A \ (h \leq 20km)$$

(17a)

$$F_n(t) = \left(M_0 - \int_0^t \left(TSFC\frac{F_n(t)}{1 - R_r p_e}\right) dt\right) \cdot (a + g) + 0.375\left(-2.2 \times 10^{-6}\int_0^t \left(\int_0^t a(t)\right) dt + 0.1331\right)\left(\int_0^t a\, dt\right)^2 A \ (20 < h < 60km)$$

(17b)

$$F_n(t) = \left(M_0 - \int_0^t \left(TSFC\frac{F_n(t)}{1 - R_r p_e}\right) dt\right) \cdot (a + g) \ (h \geq 60km)$$

(17c)

Eq. 17 cannot be solved directly because $F_n(t)$ appears on both sides of the equation. To solve Eq. 17 using a numerical integration approach, its integral format needs to be converted into a discrete form (Eq. 18), in which each time step (k starting from 0) is taken as a second in this study:

$$F_{n,k} = \left(M_0 - \sum_{i=0}^{k-1} TSFC\frac{F_{n,i}}{1 - R_r p_e}\right) * (a + g) \pm 0.375\frac{p_0 M_m}{RT_0} * \left(1 - \frac{L * h_{k-1}}{T_0}\right)^{\frac{gM_m}{RL} - 1} * (v_{k-1})^2 A \ (h \leq 20km)$$

(18a)

$$F_{n,k} = \left(M_0 - \sum_{i=0}^{k-1} TSFC\frac{F_{n,i}}{1 - R_r p_e}\right) * (a + g) \pm 0.375\left(-2.2 \times 10^{-6}h_{k-1} + 0.1331\right) * (v_{k-1})^2 A \ (20 < h < 60km)$$

(18b)

$$F_{n,k} = \left(M_0 - \sum_{i=0}^{k-1} TSFC\frac{F_{n,i}}{1 - R_r p_e}\right) * (a + g) \ (h \geq 60km)$$

(18c)

Eq. 18 needs to be integrated separately for launching and landing by flipping the direction of air drag as indicated by "±" before the coefficient of 0.375.

To solve Eq. 18, the boundary conditions need to be specified, such as:

• Initial velocity is zero;

• Final landing velocity is zero;

• The travel distance in the ascending and descending phases is 100 km, i.e., reaching the Kármán line;

• The total trip time is 10 minutes. All Blue Origin flights are 10–11 minutes in duration (https://www.blueorigin.com/new-shepard). The Virgin Galactic flights are a little longer due to their two-stage process;

• Both launching and landing involve acceleration and deceleration processes. Specifically, the acceleration and deceleration processes of launching have accelerations of a (before the engine shuts down) and g (after the engine shuts down), respectively. In contrast, the acceleration and deceleration processes of landing have accelerations of g (before the engine re-ignition) and a (after the engine re-ignition), respectively;

• The cruising velocity (i.e., the spacecraft's velocity when it reaches an altitude of 100 km) is negligible [62];

• Descending and ascending have a roughly equal amount of time.

For this study, our integration assumes an initial propellant mass. The initial total mass, $M_0$, will be the sum of the spacecraft's dry mass (a known quantity but spacecraft-specific, Section 4.1) and initial propellant mass (an assumed value to start with). For each second, the thrust will be calculated, and then the propellant consumption will be calculated. For the next time step (i.e., second), the total mass will be reduced by the amount of consumed propellant. This is repeated until the spacecraft returns to the ground. The calculated total propellant consumption should equal the initially assumed total propellant mass (assuming negligible fuel leftover). Otherwise, a new assumed initial total propellant mass will be used, and the procedure will be repeated. Specifically, if the assumed propellant is more than the total calculated consumption, in the following iteration, the assumed propellent will be reduced by half of the difference, and vice versa. The analysis was completed when the difference between the calculated and assumed propellant mass was less than 1%. Our iterative process is illustrated in the flowchart below (Fig. 2).

Our model and analysis focus on single-stage, vertical launching, which is the case for most suborbital spacecraft, including Blue Origin and SpaceX. Virgin Galactic launches its spacecraft horizontally and in two stages: in the first stage, the spacecraft is airlifted to an altitude of 14 km by WhiteKnight Two, and then the spacecraft will take off from that elevation. Dissel, Kothari [63] made a general comparison of different launching methods and concluded that for a single-stage launching, horizontally launched spacecraft needed to be 30% heavier than vertically launched ones and, thus, needed more fuels. A two-stage launching is a way to save energy because it is a way to get rid of some weight after stage one. Therefore, our calculation of Virgin Galactic may be slightly biased high.

Eq. 18 includes characteristic parameters of spacecraft, such as spacecraft dry mass and TSFC. All these key parameters are spacecraft-specific and not readily publicly available. Thus, their derivation and estimation are detailed in Section – Spacecraft-specific parameters. Once the propellant consumption for each trip is determined from Eq. 18 based on

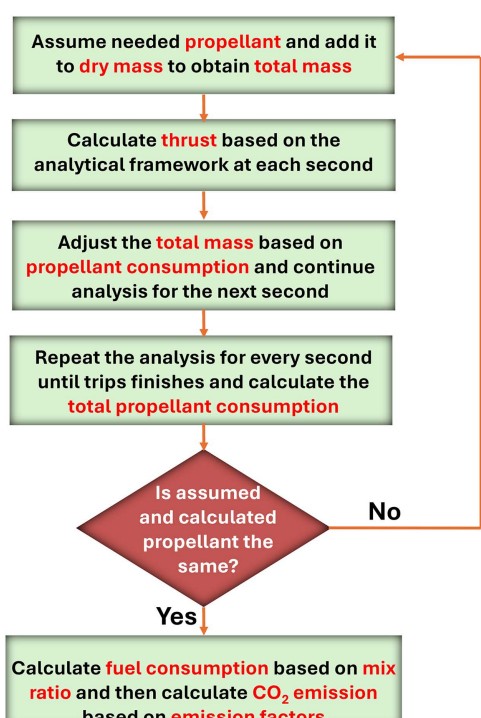

**Fig 2. Outline of the iterative process to solve the kinetics, thrust, and propellent consumption rate during the flight.** The bottom box refers to the derivation of fuel consumption and $CO_2$ emissions (details in the next section).

given dry spacecraft mass (Subsection – Spacecraft dry mass) and TSFC (Subsection TSFCs), the fuel (as part of the propellant) consumption is calculated based on the mix ratio (Subsection Mix ratios). Afterward, $CO_2$ emissions are calculated based on the emission factors (Subsection Emission factors).

### Spacecraft-specific parameters

**Spacecraft dry masses.** The total mass of a spacecraft consists of dry mass (sometimes called empty mass) and propellant mass. If the dry mass is given, the amount of needed propellant can be determined based on the analytical framework (Eqns. 6, 14, and 16). The reported dry masses of the spacecraft of Blue Origin, Virgin Galactic, and SpaceX are listed in Table 3. The dry mass of the Virgin Galactic spacecraft is much smaller than that of Blue Origin and SpaceX Dragon, which may be attributable to its launch method. Virgin Galactic uses two aircraft (i.e., WhiteKnight Two) to lift the spacecraft (i.e., VSS Unity) to about 14–16 km in the air, and then the spacecraft launches on its own engine. The dry mass of Starship is much greater than these of others because it is designed to take hundreds of passengers for commercial purposes. Therefore, to make results comparable for this study, SpaceX Falcon 9 with Dragon capsule, able to take six passengers, is used as SpaceX's spacecraft for suborbital tourism in this study.

### TSFCs

The critical parameter, TSFC, indicates the fuel efficiency of an engine design with respect to thrust output, which is an important design parameter for thrust engines, including turbojets, turbofans, ramjets, and rockets. TSFC depends on the propellant type (broadly including fuel and oxidizer) and engine design, which is deemed proprietary data since it could impact commercial competitiveness. An indirect approach is taken to estimate TSFC.

For example, the BE-3 engine of Blue Origin uses a liquid hydrogen and oxygen mixture as its propellant, and the published TSFCs of similar rocket engines using the same fuel are collected in Japan, France, and the US sources (Table 4). These engines demonstrate great consistency (the mean and standard deviation of TSFCs are 227 and 5 g/kN·s, respectively). The 10%, 50%, and 90% percentile of TSFC (220, 227, and 234, respectively) will be used to represent the central estimate and the likely range for Blue Origin BE-3 (Table 4).

Virgin Galactic's VSS Unity uses a hybrid propellant comprised of a solid fuel, hydroxyl-terminated polybutadiene (HTPB), and a liquid oxidant, nitrous oxide [73]. HTPB is a solid, rubber-like fuel made by combining two components: 1) a hydroxyl-terminated polymer of butadiene and 2) a cross-linking agent, either isocyanate or methylene diphenyl diisocyanate (MDI). The TSFC data of existing HTPB engines are collected from different sources [74,75]. If a specific impulse ($I_{sp}$) is provided in the publications, it is converted to TSFC. $I_{sp}$ is defined as thrust produced per unit of fuel consumed, which is inversely proportional to TSFC [76]. The estimated TSFC for Virgin Galactic is presented in Table 5.

SpaceX developed the Merlin family of engines to support its space ambition [77], as well as Raptors for its larger Starships [78,79]. Merlin 1C uses liquid RP-1 and $O_x$ as a propellant. RP-1, Rocket Propellant-1 or Refined Petroleum-1, is a highly refined form of kerosene [80]. The Raptor engines use liquid methane, which may also be used for other engines that SpaceX may develop. Based on the third-party test results and SpaceX's own data on the specific impulse

**Table 3. Dry mass of spacecraft [64–68].**

| Carrier | Spacecraft | Dry mass (kg) |
| --- | --- | --- |
| Blue Origin | New Shepard | 20,500 |
| Virgin Galactic | SpaceShipTwo class: VSS Unity | 6,132 |
| SpaceX | Falcon 9 + Dragon | 25,600 |
| SpaceX | Starship | ~ 85,000 |

**Table 4. TSFC of known liquid hydrogen rocket engines [69–72].**

| Company | Engine model | $H_2/O_x$ mixing ratio | TSFC (under the vacuum condition) | |
|---|---|---|---|---|
| | | | lb/lbf·h | g/kN·s |
| Mitsubishi Heavy Industries | LE-7A | 1:5.9 | 8.22 | 233 |
| Snecma (now Safran Aircraft Engines) | HM-7B | 1:5 | 8.10 | 229 |
| Mitsubishi Heavy Industries | LE-5B-2 | 1:5.5 | 8.05 | 228 |
| Rocketdyne, Pratt & Whitney Rocketdyne, Aerojet Rocketdyne | RS-25 | 1:6.03 | 7.95 | 225 |
| Aerojet Rocketdyne | RL-10B-2 | 1:5.88 | 7.73 | 219 |
| Average | | 1:5.66 | 8.01 | 227 |
| Standard deviation | | 0.014 | 0.18 | 5.2 |

**Table 5. TSFCs and mixing ratios for Big-3's engines and fuels.**

| Propellant | Engine | Carrier | TSFC (g/kN·s) | | | Mix ratio |
|---|---|---|---|---|---|---|
| | | | Upper range | Central estimate | Lower range | |
| Cryogenic ($H_2/O_x$) | BE-3 | Blue Origin | 220 | 227 | 234 | 1:5.66 |
| Solid (HTPB/$NO_x$) | VSS Unity | Virgin Galactic | 440 | 454 | 468 | 16.7:1 |
| Kerosene (RP-1/$O_x$) | Falcon's Merlin 1C | SpaceX | 323 | 333 | 343 | 1:2.17 |
| Natural gas ($CH_4/O_x$) | Starship's Raptor | SpaceX | 299 | 310 | 321 | 1:3.6 |

($I_{sp}$) of its methane engine [81–83], the TSFC was calculated to be consistent with a mean value of approximately 310 g/kN·s (Table 5).

## Mix ratios

TSFC estimates the total propellant consumption, yet the propellant is a mix of fuel and oxidizer. Based on the mix ratio, fuel consumption can be derived. Table 5 (the rightmost column) lists the collected mix ratio for various fuels. The mix ratios for $H_2/O_x$ engines were collected when the TSFCs were collected, and their references are provided in Table 4. The RP-1/$O_x$ mixture ratio is 1:2.17 according to the detailed engineering parameters of the Merlin engine family [84]. Virgin Galactic did not publish the data, but based on stoichiometric analysis of Chen, Lai [85], the mass ratio of HTPB/$NO_x$ of 16.7 is used.

## Emission factors

After deriving the total thrust, we can estimate the propellant (fuel and oxidants) mass based on TSPC, and then, using the mixing ratio, we can obtain the actual fuel weight. The last step in the derivation requires emission factors to translate fuel consumption to $CO_2$ emissions (Table 6), which is defined as a ratio of emission mass to fuel mass. $H_2$ does not directly emit $CO_2$ when burned; however, its production can release $CO_2$, which is an indirect emission. Hydrogen ($H_2$) is deemed one of the most important energy sources in the future to achieve net zero emissions. According to the International Energy Agency (IEA), the broad adoption of hydrogen as an alternative energy source could account for 6% of cumulative $CO_2$ emission reduction [86]. In 2021, 94 million tons of hydrogen were produced [87] and can be classified as "gray", "blue", and "green" based on $CO_2$ emissions [88,89]. "Gray" hydrogen production releases significant $CO_2$, e.g., via steam methane reforming (SMR) [90], which uses natural gas ($CH_4$) as feedstock to react with water under heat, giving hydrogen and carbon monoxide. However, if carbon capture and storage are used to collect and sequestrate the

**Table 6. $CO_2$ emission factors ($CO_2$ vs. fuel) of different fuels [94–96].**

| | Direct emissions due to Combustion | Indirect emissions due to Production | |
| --- | --- | --- | --- |
| | Current and Future (g/g) | Current (mid-2020s) (g/g) | Future (2030s) (g/g) |
| Hydrogen | 0 | 10.1–11.8 | 6.7–7.6 |
| HTPB | 2.2–2.6* | ---- | ---- |
| RP-1 | 3.16 - 3.5 | 0.5 - 2.1 | 0.4 - 1.5 |
| $CH_4$ | 2.6 - 2.75 | 0.13–0.44 | 0.1 - 0.18 |

Note: *the emission factor range for HTPB is calculated from its theoretical emissions from combustion and then considering engine efficiency.

produced carbon monoxide (and $CO_2$), the "gray" hydrogen turns into "blue" hydrogen [91]. In contrast, "green" hydrogen is produced with no $CO_2$ emissions [92], for example, utilizing thermochemical or electrical processes to achieve water splitting with solar, wind, or hydropower energy. So far, "gray" hydrogen accounts for more than 95% of the supply in the global market due to its high yield and low cost [93]. The United States produces 9–10 million tons of hydrogen annually, nearly 99% from SMR. However, based on IEA's projection, "blue" or "green" hydrogen will account for more than 95% of the market by 2050. Therefore, even with the total production increasing by more than four times, $CO_2$ emissions can still slightly decrease in 2050. According to IEA. [86], $CO_2$ emissions associated with producing 1 kg of $H_2$ ranged from 12 to 13.5 kg in 2022 and will gradually decrease to 6.7 to 7.6 kg in the 2030s due to the increasing use of green hydrogen [86]. For this study, based on a linear interpretation of IEA data, the emission factors of 10.1–11.8 and 6.7–7.6 will be used as emission factors for hydrogen in the 2020s (current) and 2030s, respectively (Table 6).

Direct emissions are associated with combustion of all carbon-containing fuels (HTPB, RP-1, and $CH_4$). For $CO_2$ emissions due to HTPB fuels, stoichiometry was used to calculate the emission factors based on the chemical structure presented in Khan, Abhijit Dey [97]. The 60% carbon content in the HTPB molecule serves as the basis for the calculation of $CO_2$ emissions. Due to different crosslink agents used in polymerization, the molecules of HTPB and the carbon content may differ slightly. HTPB can be produced from oligomers from different chemical pathways [98,99]. There is no information regarding the $CO_2$ emissions of HTPB production, so that part is omitted here.

The $CO_2$ emissions associated with RP-1 fuel are estimated based on US EPA emission factor. The combustion of each gallon of kerosene would directly release 9.98 kg of $CO_2$ [100], which is converted to a mass ratio, i.e., emission factor. Koroneos, Dompros [101] and GHG Management Institute [95] indicated the emission factors associated with kerosene production range from 0.5 to 2.1, which are listed in Table 6.

Detailed data on $CO_2$ emissions associated with natural gas production is not available, but Zhang, Cusworth [96] concluded that natural gas liquefaction released only 5–16% $CO_2$ relative to end-use combustion. Okamura, Furukawa and Tamura, Tanaka reported that 230–410 g of $CO_2$ is released to liquefy each kilogram of natural gas [102,103], which is equivalent to 8.5% to 15.2% of end-use emissions. Howarth suggested that liquefaction accounts for 8.8% $CO_2$ emissions relative to end-use based on the U.S. data [104]. Thus, it is supportive to use 5–16% as the lower and upper bound of $CO_2$ emissions due to liquefaction. It is noteworthy that at this moment, SpaceX is using natural gas for its raptor engines, but Blue Origin indicated that its future BE-4 engine will use liquefied natural gas as well. That means natural gas engines may become more popular in the future.

In addition to the current indirect emission factor, the future projection for the 2030s is also listed in Table 6. The prediction is based on the U.S.'s target to increase renewable energy to 40% of total energy used, which is the average target of different states' legislature [105]. Even though a lot of countries plan to increase their renewable energy to 80% or even 90% by 2030s, such as Australia and Malaysia, 40% is widely used by many other countries as a renewable energy target, such as the European Union and the United Kingdom [106]. As a result, fuel production will release less $CO_2$. The 40% renewable energy target is also largely consistent with the emission factor for hydrogen in the 2030s.

## Monte Carlo analysis

The calculation of emissions involves two parameters – fuel consumption and emission factors. Since the TSFC of the fuels varies within a range (listed in Table 5), the calculated fuel consumption will fall into a range. The emissions factors (listed in Table 6) also fall into a range. Therefore, a Monte Carlo simulation, which is a computational technique that uses repeated random sampling to obtain numerical results, is utilized to estimate the emissions considering the combination of various scenarios of TSFCs and emission factors. Both TSFCs and emission factors are assumed to fall into a normal distribution, and the lower and upper bounds of the distributions are assumed to correspond to 10% and 90% percentile, respectively. To ensure accuracy and convergence, the simulation starts with 2,000 cycles and is increased by 500 cycles each time until the yielded mean and standard deviation become nearly unchanged. The Monte Carlo simulation is used to quantify not only $CO_2$ emissions from fuel combustion or production but also the total $CO_2$ emissions, i.e., emissions from combustion + production.

All the collected and derived information, developed analytical framework, and adopted Monte Carlo analysis are incorporated into the analysis to link fuel consumption to total $CO_2$ emissions, as shown in Fig. 3.

## Results and discussions

Based on the analytical framework developed (Section 3) and extensive data collected from various sources (Section 4 on spacecraft characteristics, combustion efficiency, emission factors, fuel production, etc.), we can calculate fuel consumption and, consequently, $CO_2$ emissions (see the workflow in Fig. 3) of the dominant carriers of commercial suborbital tourism (Section 5.1). We also estimate the future emissions based on the projected demand and potential improvement of emission factors (Section 5.2).

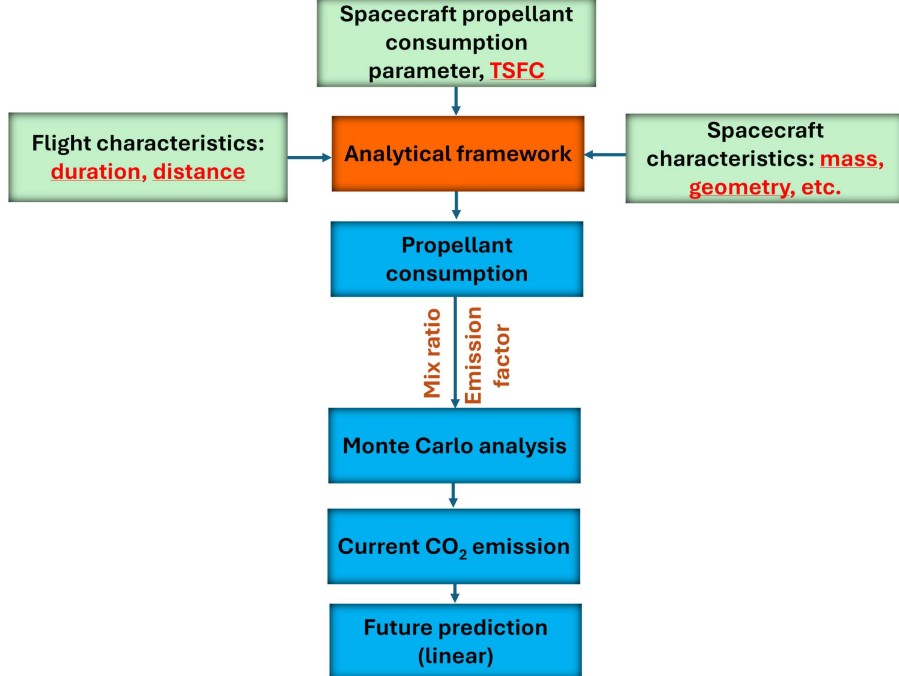

**Fig. 3. The framework of estimating fuel consumption and $CO_2$ emissions.**

## Estimates of current emissions

The calculated propellant, fuel consumption, and $CO_2$ emissions follow a normal distribution, as shown in Fig. 4a, due to the variation of TSFCs and emissions factors. To avoid displaying a normal distribution for each calculated value, an error bar in the charts indicates the 10% − 90% range. The total needed propellants (fuel and oxidizer) for each trip from different carriers are presented in Fig. 4b, compared with the dry masses of the spacecraft. It is evident the total propellant mass significantly surpasses the dry masses of spacecraft. Among them, Blue Origin carries the least amount of propellant compared with others. This is understandable as hydrogen has the highest chemical energy density of 121 MJ/kg (the amount of energy released for burning a unit amount of fuel). In contrast, methane is 55 MJ/kg, and kerosene is 43 MJ/kg. As a result, it can reduce the total mass of the spacecraft, which would further reduce the propellant needed. This is also consistent with the fact that Blue Origin has lower TSFC than others (Table 5).

$CO_2$ emissions associated with one trip of suborbital flight for each carrier are presented in Table 7 and the emissions from combustion and production are shown in Fig. 4c. We find that natural gas ($CH_4$) is the one that results in the least total $CO_2$ emissions at 85 tons per trip, despite the largest dry mass of spacecraft at 25,600 kg (Table 3). Surprisingly, liquid hydrogen, without direct emissions, has slightly more $CO_2$ emissions (98 tons per trip) than natural gas due to its production. HTPB and RP-1 are significantly higher at 129 and 226 tons per trip, respectively. Notably, the $CO_2$ associated with RP-1 is a few times that of others. Due to a lack of information, we did not include $CO_2$ emissions associated with HTPB production, which would cause some underestimation of $CO_2$ emissions of HTPB.

SpaceX, using RP-1 or natural gas, has a greater total mass due to their higher TSFCs, which means less thrust is provided by the unit mass of fuel consumption. Liquid hydrogen has low TSFC, so the total mass of Blue Origin spacecraft is low. The Virgin Galactic spacecraft's total mass is low because the dry weight is low, thanks to its two-stage launching system. The reported total mass of Blue Origin NS-13 (i.e., dry mass + fuel mass at launching) is 75 tons, while our calculation is 79 tons (within a 5% difference). Unfortunately, other carriers' data is not directly comparable: the current SpaceX systems go further than suborbital, and Virgin Galactic uses a two-stage launch.

Next, we show the details of the spacecraft's mass changes during flights based on our analytical model (Fig. 5a). The initial total mass (at t = 0) includes the spacecraft's dry mass and total propellant mass. Therefore, the decrease in total mass is the consumption of propellant, and at the end of the flight, the remaining is the spacecraft's dry mass. In companion with the spacecraft's mass decrease, $CO_2$ is released into the atmosphere during the flight. Fig. 5b shows the cumulative $CO_2$ emissions with respect to flight time. Any $CO_2$ emissions associated with production are concentrated at t = 0 sec in the figure. The middle plateaus of the curves indicate the engine shutdown, which is consistent with no mass change in the center portion of the curves in Fig. 5a.

The $CO_2$ emissions at different altitudes and times during the flight are presented in Fig. 6, which is only due to combustion. Fig. 6a separates $CO_2$ emission into ascending and descending phases while Fig. 6b combines them. The spacecraft goes to an altitude of 100 km, but the engine shuts down at an altitude of nearly 80 km, so no $CO_2$ emissions occur beyond that point. As seen in Fig. 6a, most $CO_2$ emissions occur at the ascending phase, and the total emissions at the descending phase are only approximately ¼ of that of the ascending phases. This analysis is based on a controlled landing scenario without a parachute; thus, could be overestimating Blue Origin (crew capsule landing on parachute), and VG (horizontal landing). Nevertheless, while this measure to increase drag can be used to reduce fuel burning in landing, it would not have a great impact on the total $CO_2$ emissions as most of them occur at the ascending phase. Comparing $CO_2$ emissions from ascending and descending phases, curves of $CO_2$ emissions in descending show a non-monotonic trend, which is attributable to air drag. In the ascending phase, air drag acts as a force against acceleration, requiring the spacecraft to generate more thrust to overcome it. In contrast, in the descending phase, air drag helps decelerate spacecraft, so less thrust is needed, particularly, below 20 km, where air becomes thicker and thicker. Fig. 6b, combining emissions from ascending and descending phases, can better show $CO_2$ emissions as a function of altitude. Most emissions occur at low

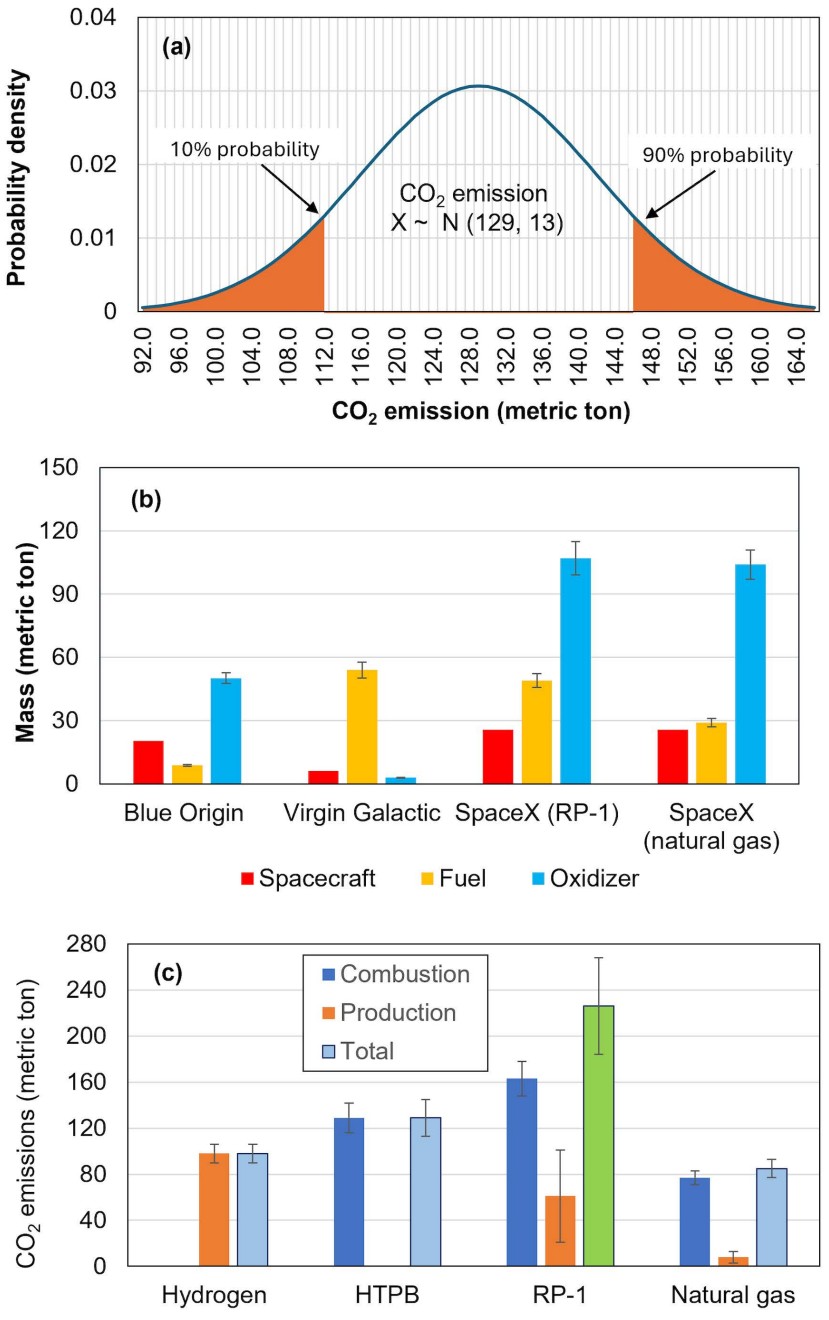

**Fig. 4. Propellant and CO$_2$ emissions per trip:** (a) CO$_2$ emission distribution of HTPB, (b) mass of spacecraft, fuel, and oxidizer, and (c) CO$_2$ emissions due to combustion and production.

altitudes. The emissions at the first 20 km represent nearly 60% of the total emissions. Further study is needed to investigate the differential effect of emissions at various altitudes.

Let us now put the CO$_2$ emissions in Table 7 into context. There are various estimates of CO$_2$ associated with air travel on commercial aircraft. Typically, a transatlantic flight would emit 620–1200 kg of CO$_2$ per passenger per round trip [107],

**Table 7. Propellant and fuel consumptions and variations.**

| | Mean/Standard deviation | | | |
|---|---|---|---|---|
| **Carrier (Fuel/Oxidizer)** | **Blue origin ($H_2/O_x$)** | **Virgin Galactic (HTPB/NOx)** | **SpaceX (RP-1/$O_x$)** | **SpaceX ($CH_4/O_x$)** |
| Propellant (ton) | 59/3 | 57/4 | 156/11 | 133/9 |
| Fuel (ton) | 8.9/0.46 | 54/3.9 | 49/3.3 | 29/2.0 |
| $CO_2$ emissions (combustion) (ton) | 0 | 129/13 | 163/15 | 77/6 |
| $CO_2$ emissions (production) (ton) | 98/8 | n/a | 63/31 | 8/4 |
| Total $CO_2$ emissions (ton) | 98/8 | 129/13 | 226/42 | 85/8 |
| Emissions per passenger per hour (ton) | 98/8 | 129/13 | 226/42 | 85/8 |

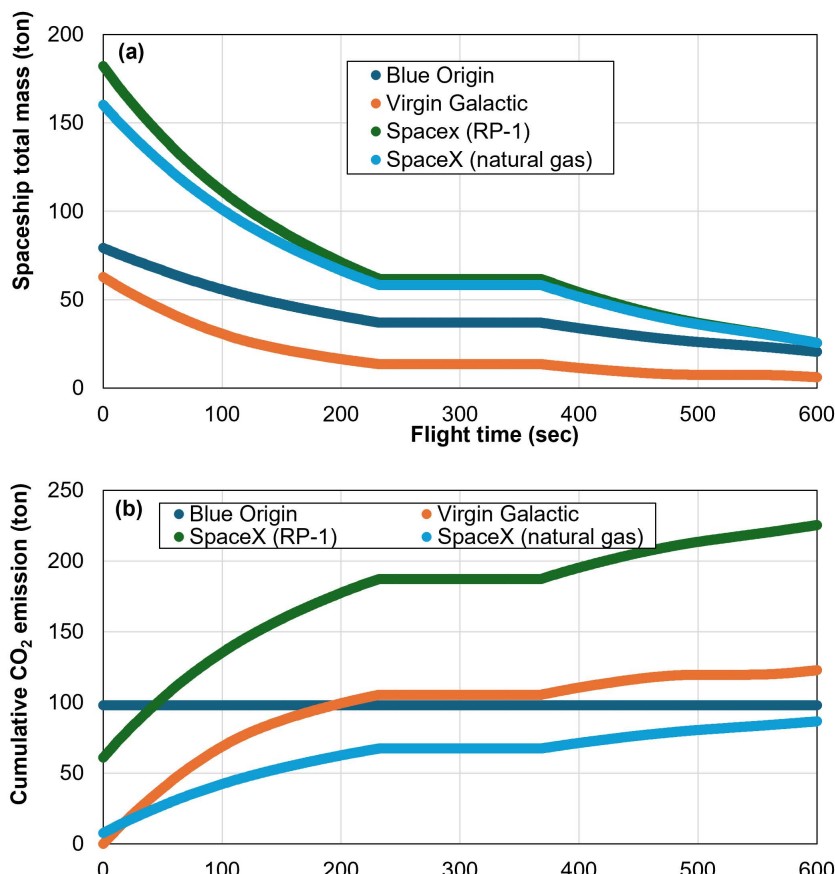

**Fig. 5. Total spacecraft mass and $CO_2$ emissions during flights: (a) mass variation; and (b) cumulative $CO_2$ emissions.**

while smaller airplanes traveling a shorter distance would emit much less, which is about 250 kg per passenger per round trip [108]. Considering the difference in flight duration and aircraft fuel efficiency, it is generally considered that the $CO_2$ emissions range from 134 to 250 kg per passenger per hour of flight [109,110].

When the total emissions per hour of a suborbital trip are converted to $CO_2$ emissions per passenger per hour of flight, $CO_2$ emissions per person per hour is shown in the rightmost column of Table 7. Note that the total emissions were converted to "Emissions per passenger per hour" based on a 10-minute flight that carries 6 passengers. The current Blue

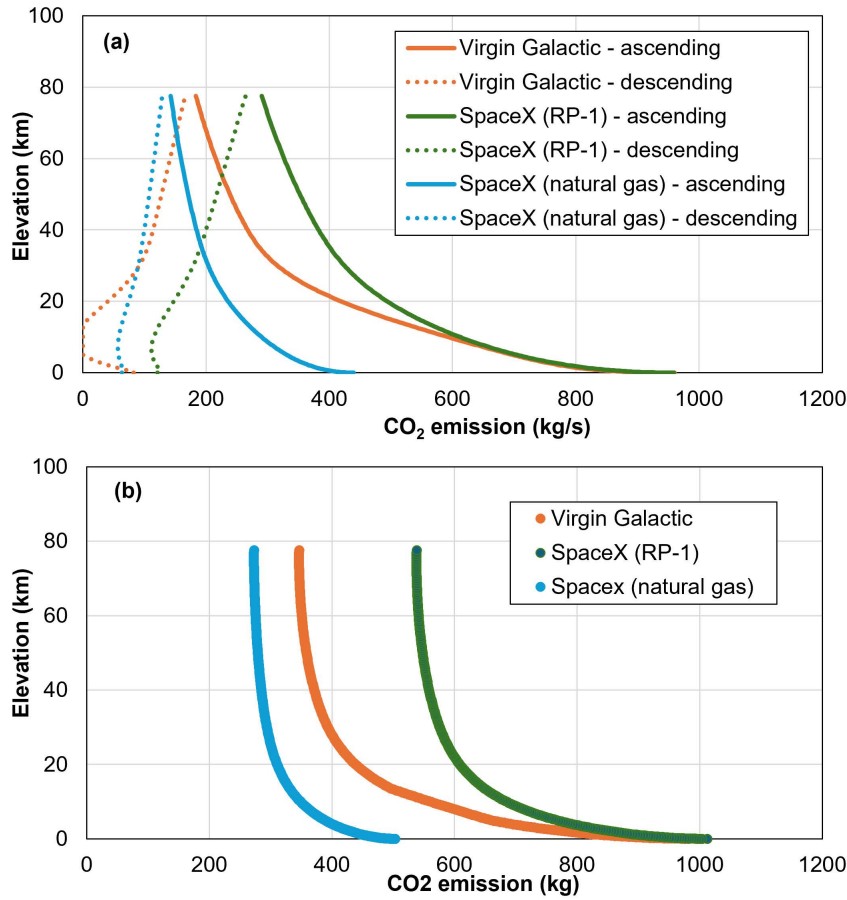

**Fig. 6.** $CO_2$ **emissions at different altitudes: (a)** $CO_2$ **emissions for ascending and descending phases at different altitudes; and (b) total** $CO_2$ **emissions at different altitudes.**

Origin commercial flight used one 10-minute duration. The 6-passenger capsules have been used for all Big-3 carriers. In any case, suborbital tourism of three carriers will be 400–1,000 times that of commercial air travel per passenger per hour. Even though Starship is not the focus of this study, we can expect that it will release a lot more total $CO_2$ because it is much heavier. However, because of the larger number of passengers on board (it has been reported it can take 200–1,000 people) and its launch method, the $CO_2$ emissions per passenger of Starship need to be carefully assessed in a future study.

One may argue that it is not fair to compare suborbital tourism with air travel because the former is mostly for sight-seeing while the latter is mostly for transportation. Next, we compare the emissions of suborbital tourism with helicopter tourism (similarly landing at the site of launches), which is another fast-growing industry. In 2022, the helicopter sightseeing market was 658 million dollars and was expected to reach nearly 1 billion dollars before 2027 [111]. The capacities of helicopters vary very much. Light single-engine helicopters can lift no more than 3 passengers, for example, the Robinson R44, which is ideal for short-distance trips. Medium helicopters, for example, Bell 505 Airbus H125, Bell 206, Leonardo AW109 Trekker, Airbus H130, MD Helicopters MD 500E, can take 4–8 passengers. Heavy helicopters, such as Sikorsky S-92, Sikorsky S-76D, AW139, Airbus H155, Bell 525 Relentless, can take more than 10 passengers. The fuel consumption varies depending on the engine and the number of passengers. Assuming jet fuel was used by them and using

medium helicopters as a representative, Hjalmarsson [112] indicated that Bell 206 would need 75 kg of fuel for a one-hour flight. Using the emission factors to account for emissions from combustion and production, we converted fuel into 420 kg of $CO_2$ emissions per hour, which is 84 kg per passenger per hour. The results indicated that $CO_2$ associated with suborbital travel is at least 1,000 times the emissions associated with medium helicopters per passenger per hour.

## Future estimates

As demonstrated in Section 5.1, suborbital tourism is highly emission-intensive and deserves extensive attention. The propellants used by different carriers greatly impact the source of $CO_2$ emissions (Fig. 4b). Different $CO_2$ sources also determine their potential for $CO_2$ reduction in the future. The $CO_2$ emissions from combustion cannot be reduced and sequestrated because it is needed to create high-velocity gas flow during flight, which is the source of propulsion. However, $CO_2$ emissions from production can be reduced by using renewable energy or sequestration. All the associated $CO_2$ emissions for hydrogen come from production, as its combustion only releases water. In contrast, for RP-1, both combustion and production release a significant amount of $CO_2$. Although natural gas is best in terms of total $CO_2$ emissions at this moment, it has limited potential for reduction as its $CO_2$ is primarily from combustion (90%). In the long term, liquid hydrogen appears to have the greatest potential in future decades to reduce $CO_2$ emissions, as discussed next.

Fig. 7 shows conceivable $CO_2$ emissions per flight in the next decade compared to the present. As discussed previously, $CO_2$ from combustion during flights is intrinsic and cannot be reduced, so all the reduction comes from the production side. $CO_2$ emissions from liquid hydrogen production depend on not only the energy consumed in production but also the raw materials used to produce hydrogen. $CO_2$ emissions from liquid hydrogen production can be reduced based on the projected future emissions factor by IEA. [86]. For RP-1 and natural gas, the $CO_2$ emissions are calculated based on the plan of 40% renewable energy by 2030, which is a target by many countries such as the U.S., European Union, and United Kingdom, which has been discussed in Section 4.4. Based on Fig. 7, $CO_2$ emissions from all carriers per flight will decrease but by different degrees. Liquid hydrogen will decrease the most, and we expect that the decrease could continue toward the late 21st century. HTPB shows no reduction potential in the future in the figure because the estimate only includes emissions from combustion.

Decision-making should consider not only the existing emissions but also account for the future growth trend [113–115]. Even though the impact of emissions from a suborbital flight is insignificant compared to current commercial aviation, its growth trend and potential environmental impacts are concerning [73]. Virgin Galactic alone has vowed to launch 400 flights each year [116]. The impact will be significant if the annual launches increase to hundreds or thousands [12,40,117].

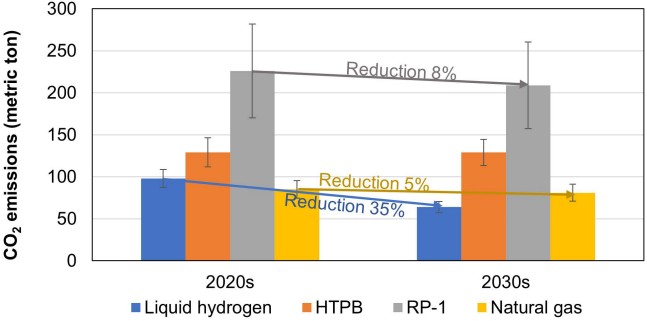

**Fig. 7. The trend of $CO_2$ emissions per flight for different fuels used in suborbital spacecraft.**

Since each suborbital trip releases a great amount of $CO_2$, the total annual release of $CO_2$ from suborbital flight can be a concern. Survey data indicated that about 80% of people between the ages of 20 and 29 were considering space travel [31]. The Pew Research Center studies showed that 42% of Americans definitely or probably were interested in space tourism [118]. It is expected most of them will be suborbital travel. Among the 52% who indicated they were not interested, 28% cited the high price as the major reason for choosing "No". The price could significantly drop if more and more people take the trip. A good example is the cost of commercial aviation. Since 1980, the cost of air travel per mile has been reduced by 50% after adjusting for inflation [119]. Therefore, the acceptance of suborbital travel can be significantly higher than indicated by the survey data. Virgin Galactic alone has vowed to launch 400 flights each year [116], and as of June 2024, 800 reservations have been confirmed per Virgin Galactic's quarterly earnings reports (https://www.sec.gov/edgar/browse/?CIK=1706946).

Based on a study that considered pricing, acceptance, and technology development, the annual number of passengers on suborbital flights could be as high as 85,000 by the 2030s [120]. Moreover, BIS Research [121] predicted a nearly 25% compound annual growth rate (CAGR) for the suborbital tourism market, and other resources predicted greater than 30% or greater than 40% CAGR [122]. If that growth rate remains, the suborbital flights will double every three years. Using the speculated 85,000 projection number, the total $CO_2$ emissions will be as high as 7–21 megatons per year in the 2030s. If the high-end number is used (i.e., 21 megatons), it will put the suborbital industry equal to how much **Bolivia** emitted in 2023 (population 12.24 million, ranked 84th among 206 countries in terms of annual $CO_2$ emissions). Even if we use the low-end number, it will still be approximately same as **Congo** (population 106 million, ranked 124th in terms of emissions). This raises numerous issues related to equity and climate justice that the public needs to consider.

The long-term implications of $CO_2$ emissions from suborbital tourism are broad and impactful, considering that it is still at the early stage of adoption. We even consider that 85,000 launches per year in 2030s would be a small portion when the market becomes mature. Logistic function has been widely used by market researchers to predict the public acceptance of newly emerging technology, which has successfully predicted the market penetration of cell phones, personal computers, smart phones, electrical vehicles, etc. [123]. A genetic logistic function is divided into three phases based on its market penetration: early adoption (<10% market penetration), wide acceptance (10%−90% market penetration), and maturation (>90% market penetration). In the early adoption phase, the sales growth is slow, while in the wide acceptance phase the growth is rapid and exponential [124]. Considering that 85,000 flights are still significantly less than 10% of total addressable market (TAM) of suborbital tourism, we can infer that suborbital tourism will be still at the early adoption phase. Smart phones took about 7 years to enter the second phase, while personal computers took slightly longer than 10 years. Thus, using that analogy suborbital tourism may enter the wide acceptance phase after mid-2030, and by then the annual launches could be a few times more than 85,000 flights, which can really impose a challenge due to its possible $CO_2$ emissions. More detailed analysis could be possible in the future when more market data on suborbital tourism becomes available.

According to this study, the key factors impacting emissions are engine efficiency (i.e., TSFC), types of fuel (i.e., emissions factors and fuel mass), and spacecraft mass. Improving spacecraft mass and engine efficiency is challenging unless a revolutionary breakthrough occurs in the near future. The choice of fuel can significantly affect emissions, and our study shows that in the long term liquid hydrogen and natural gas are better choice than other fuels so they should be prioritized to mitigate $CO_2$ emissions associated with suborbital tourism.

## Conclusions

In addition to many environmental concerns, a significant discrepancy in $CO_2$ emissions from suborbital tourism exists due to a lack of data availability. This study achieves three objectives: [1] developing a simple and sound analytical framework to analyze single-stage, vertical launches rocket to fill the gas of existing studies, [2] linking the developed framework with suborbital flight parameters and variables to estimate $CO_2$ emissions, and [3] provided comprehensive Monte Carlo simulations of different scenarios. The developed analytical model to quantify energy consumption is based on flight kinetics

and then extensively gathers information from different sources on spacecraft characteristics, engine performance, propellant mix ratio, $CO_2$ emissions from production and combustion. The analytical framework is flexible and can be refined in the future as more data becomes available. For example, the developed analytical model can be used to analyze the spacecraft of other carriers as long as the key parameters are known.

Here, we estimate that the current emission of suborbital flights ranges from 85 to 226 tons per trip, with the spacecraft using liquid hydrogen or natural gas being the least polluting. It appears that traditional aviation fuels RP-1 and HTPB release more than newly emerging fuels, i.e., liquid hydrogen and natural gas. $CO_2$ emissions associated with RP-1 are nearly three times that from hydrogen and natural gas when combustion and production are both considered. For HTPB, $CO_2$ from combustion alone is already approximately 20% − 40% higher than those from production and combustion for liquid hydrogen and natural gas. Thus, HTPB should be considered a high-emission fuel.

The emissions range from 85 to 226 tons per person and per hour, which is 400–1,000 times that of commercial flights and helicopter sightseeing. For example, the associated $CO_2$ emissions for Blue Origin's NS (New Shepard) is about 98 tons per passenger per hour, which is significantly higher than that of commercial flight passengers, which is roughly 0.2 tons per passenger per hour [109]. Considering the reported emissions from private jets, i.e., 7–9 times of commercial flight [20], $CO_2$ emissions from suborbital tourism is much higher. Considering the potential growth of suborbital tourism, we estimate that the total emissions could range from 7 to 21 Mt ton per year, which is equivalent to the total emissions by Congo and Bolivia, which were ranked 124th and 84th among 206 counties in 2023 in terms of $CO_2$ emissions.

Liquid hydrogen cannot be considered clean energy at this moment as it is primarily produced using fossil fuels. If hydrogen production moves more towards blue and green production, $CO_2$ emissions will be reduced significantly in future decades. This can, in principle, reach zero-emissions spaceflight, but it does not account for many other indirect emissions, such as the leakage of $H_2$. Even though Virgin Galactic does not release significantly more $CO_2$ compared with others, it has little potential for future reduction. Also of note is that our estimate for Virgin Galactic does not include indirect emissions from fuel production; therefore, the estimation can be refined once more fuel production information becomes available. SpaceX primarily uses RP-1 as its fuel in Falcon's Merlin engines. If it decides to enter the suborbital tourism business, it will have the highest emissions per passenger per hour compared with others, which is almost three times that of liquid hydrogen. If SpaceX decides to use natural gas engines (Raptor or less powerful ones), it will release the least $CO_2$ (85 tons per passenger per hour) at this moment, nearly comparable with liquid hydrogen. Also note that $CH_4$ can, in principle, be synthesized using green $H_2$ and captured $CO_2$ (so-called synthetic fuel or e-fuel), greatly reducing its emissions factor. The results from this study indicate that the choice of fuel will really make a difference in terms of potential $CO_2$ emissions. Suborbital tourism is at its early adoption stage and could enter the wide acceptance stage after the mid-2030s, which then will impose a significant challenge if the issue is not addressed appropriately.

In summary, our study shows that suborbital tourism will become an important contributor to $CO_2$ emissions impacting the global environment; this is in addition to other environmental concerns related to non-$CO_2$ emissions, noise, local soil and water pollution, etc. This study focuses on emissions related to fuel production and combustion. However, compared to commercial airplanes, suborbital spacecrafts are much less usable. At this moment, SpaceX and Blue Origin aim at reusing 10 or 20 times, while a commercial airplane is typically designed for 30,000 cycles and can last for 25–30 years. Thus, a holistic life cycle analysis to account for $CO_2$ emissions related to spacecraft manufacture and maintenance will be needed.

## Supporting information

**S1 File. Paper data.**
(XLS)

## Author contributions

**Conceptualization:** Yangyang Xu.

**Formal analysis:** Angela M. Huang.

**Methodology:** Angela M. Huang.

**Supervision:** Yangyang Xu.

**Writing – original draft:** Angela M. Huang.

**Writing – review & editing:** Angela M. Huang, Yangyang Xu.

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
