## [Decision Letter · Decision Letter 0]

25 Feb 2025

Dear Dr. Xu,

Thank you for submitting your manuscript to PLOS ONE. After careful consideration, we feel that it has merit but does not fully meet PLOS ONE’s publication criteria as it currently stands. Therefore, we invite you to submit a revised version of the manuscript that addresses the points raised during the review process.

We look forward to receiving your revised manuscript.

Kind regards,

Mehmet Cem Catalbas, Ph.D.

Academic Editor

PLOS ONE

Journal Requirements:

2. We note that your Data Availability Statement is currently as follows: “All relevant data are within the manuscript and in Supporting Information files.”

3. We note that Figure 1 in your submission contain copyrighted images. All PLOS content is published under the Creative Commons Attribution License (CC BY 4.0), which means that the manuscript, images, and Supporting Information files will be freely available online, and any third party is permitted to access, download, copy, distribute, and use these materials in any way, even commercially, with proper attribution. For more information, see our copyright guidelines: http://journals.plos.org/plosone/s/licenses-and-copyright.

4. We note you have included a table to which you do not refer in the text of your manuscript. Please ensure that you refer to Table 3 in your text; if accepted, production will need this reference to link the reader to the Table.

Reviewers' comments:

Reviewer's Responses to Questions

**Comments to the Author**

1. Is the manuscript technically sound, and do the data support the conclusions?

Reviewer #1: Partly

Reviewer #2: Partly

2. Has the statistical analysis been performed appropriately and rigorously?

Reviewer #1: I Don't Know

Reviewer #2: Yes

3. Have the authors made all data underlying the findings in their manuscript fully available?

Reviewer #1: Yes

Reviewer #2: Yes

4. Is the manuscript presented in an intelligible fashion and written in standard English?

Reviewer #1: Yes

Reviewer #2: No

Reviewer #1: While the article presents an interesting and relevant topic regarding CO2 emissions associated with suborbital flights, there are several aspects that raise concerns about its suitability for publication in this prestigious journal.

Firstly, the paper's approach seems somewhat elementary and may lack the depth expected for a high-impact journal. The analysis is based on basic physical principles and fundamental kinetics of launching and landing, which, although informative, may not offer the level of novelty or advanced methodology required to stand out in the field. The models proposed, while valuable, appear to be quite basic and do not seem to introduce any groundbreaking insights or advancements in the methodology used for CO2 estimation.

Furthermore, while the results are intriguing—particularly the comparison of emissions from different spacecraft and fuels—the analysis remains at a level that may be more suited to a classroom or introductory research project rather than a comprehensive scientific study. The study provides useful data but does not seem to fully explore or address the broader implications of these findings in the context of global environmental policy or the technological innovations needed to mitigate such emissions.

Additionally, while the potential scale of emissions from suborbital flights is highlighted, the article does not provide sufficient data or a detailed discussion on how these emissions could be reduced or managed. Given the growing concern around climate change and CO2 emissions, this is a crucial aspect that would enhance the paper's contribution to the field.

In conclusion, while the study has potential, it would benefit from a deeper analysis, more advanced modeling, and a clearer connection to the broader implications of suborbital flight emissions. As it stands, it may not meet the high standards expected by this journal, and I would recommend reconsideration after further revisions and improvements.

Reviewer #2: The paper by Huang et al. “Estimating CO2 Emissions Due to Present and Future Suborbital Space Tourism Industry” discussed a hot debated topic after several high-profile suborbital launches in recent years. By using physics and thermodynamics principles, the authors developed an analytical framework with governing equations to estimate the space tourism industry’s carbon footprint. I have several concerns about their methods and approximations.

The formula 3 doesn’t hold outside of the troposphere. In stratosphere (20-50 km), temperature increases with altitude. Then in the mesosphere (50-100 km), temperature decreases again. But since the authors are studying the emission from launch to 100km, equation 3 only applies for certain segments.

L28 can quantitively specify the numbers.

There are grammar errors in the draft. Please proofread.

L169 Considering drag coefficient can change with rocket shape and speed, using a constant value is way too simplistic. The authors should at least identify the main stages during the trip, and use a customized drag coefficient for each, or replace Cd using a function of flight conditions at least.

Table 2 it is interesting to see the exhaust velocity difference using liquid hydrogen and various oxidizers is all around 14%. But since the main fuel is the same (~hydrogen) in this case, oxidizers just play the role of oxidizers, so maybe that all makes sense. But what the authors did not do is for different fuels, the 14% factor might change.

Table 6 First need to list units on the table. I would also suggest the EF for hydrogen is highly uncertain while the authors assumed a linear trend and used a simple approximated value. At least some sensitivity analysis needs to be done to suggest the full range with uncertainty attached.

L443 The authors can also try to dismiss the carbon footprint estimates for HTPB since the results will be quite uncertain already.

L453 Before natural gas liquefaction, there can be flaring, combustion on sites for compression, these all lead to CO2 emissions while the authors only considered the liquefaction step. The use of 10% will be an underestimate and the authors need to find more data resources to constrain the upper bound.

L499 I doubt other columns will have similar range of uncertainty considering the large uncertainty in emission factors and other relevant variables. The other thing the authors should improve is to be more quantitative. They can simply add uncertainty ranges to all columns.

**Do you want your identity to be public for this peer review?** For information about this choice, including consent withdrawal, please see our Privacy Policy

Reviewer #1: No

Reviewer #2: No

---

## [Author Response · Author response to Decision Letter 1]

16 Apr 2025

A detailed response has been submitted along with the revised manuscript.

---

## [Decision Letter · Decision Letter 1]

12 May 2025

Dear Dr. Xu,

Thank you for submitting your manuscript to PLOS ONE. After careful consideration, we feel that it has merit but does not fully meet PLOS ONE’s publication criteria as it currently stands. Therefore, we invite you to submit a revised version of the manuscript that addresses the points raised during the review process.

We look forward to receiving your revised manuscript.

Kind regards,

Mosab Wael Alrashed

Academic Editor

PLOS ONE

Journal Requirements:

**Additional Editor Comments:**

The comments of the reviewer are as follows:

I appreciate your efforts to address the majority of my inquiries. My final apprehension pertains to the disparity in air density at the 20 km mark. I discovered that the air density at 20 km is approximately 7% of the density at sea level, which is not the 0.02% that the authors had previously identified. Consequently, it is not negligible, particularly when studying high-speed flying objects. The authors must revise their segment calculations into at least three bins: 0-20, 20-60 (above 60 km, it is safe to assume density to be zero), and 60-100 km.

Reviewers' comments:

Reviewer's Responses to Questions

**Comments to the Author**

Reviewer #2: (No Response)

2. Is the manuscript technically sound, and do the data support the conclusions?

Reviewer #2: Yes

3. Has the statistical analysis been performed appropriately and rigorously?

Reviewer #2: Yes

4. Have the authors made all data underlying the findings in their manuscript fully available?

Reviewer #2: Yes

5. Is the manuscript presented in an intelligible fashion and written in standard English?

Reviewer #2: Yes

Reviewer #2: Thanks for addressing most of my comments. My last concern is on the air density difference on the 20 km mark. What I found out is air density at 20 km is about 7% of density on sea level, not the 0.02% the authors identified, so it is not negligible especially we are studying high speed flying object. The authors need to revise their segment calculations into at least 3 bins, 0-20, 20-60 (above 60 km, seems we can safely assume density to be zero), 60 -100 km.

**Do you want your identity to be public for this peer review?** For information about this choice, including consent withdrawal, please see our Privacy Policy

Reviewer #2: No

---

## [Author Response · Author response to Decision Letter 2]

17 Jun 2025

A detailed response has been uploaded as a separate file.

---

## [Editor Report · Decision Letter 2]

2 Jul 2025

Estimating CO2 Emissions Due to Present and Future Suborbital Space Tourism Industry

PONE-D-25-00871R2

Dear Dr. Xu,

We’re pleased to inform you that your manuscript has been judged scientifically suitable for publication and will be formally accepted for publication once it meets all outstanding technical requirements.

Kind regards,

Mosab Wael Alrashed

Academic Editor

PLOS ONE
---

## [Editor Report · Acceptance letter]

PONE-D-25-00871R2

PLOS ONE

Dear Dr. Xu,

I'm pleased to inform you that your manuscript has been deemed suitable for publication in PLOS ONE. Congratulations! Your manuscript is now being handed over to our production team.

Kind regards,

on behalf of

Dr. Mosab Wael Alrashed

Academic Editor

PLOS ONE